# HYPERCORE: CORESET SELECTION UNDER NOISE VIA HYPERSPHERE MODELS

## ABSTRACT

The goal of coreset selection methods is to identify representative subsets of datasets for efficient model training. Yet, existing methods often ignore the possibility of annotation errors and require fixed pruning ratios, making them impractical in real-world settings. We present HyperCore, a robust and adaptive coreset selection framework designed explicitly for noisy environments. HyperCore leverages lightweight hypersphere models learned per class, embedding in-class samples close to a hypersphere center while naturally segregating out-of-class samples based on their distance. By using Youden's J statistic, HyperCore can adaptively select pruning thresholds, enabling automatic, noise-aware data pruning without hyperparameter tuning. Our experiments reveal that HyperCore consistently surpasses state-of-the-art coreset selection methods, especially under noisy and low-data regimes. HyperCore effectively discards mislabeled and ambiguous points, yielding compact yet highly informative subsets suitable for scalable and noise-free learning. The code for HyperCore will be published upon acceptance.

## 1 INTRODUCTION

Modern deep learning excels with scale, but scale comes at a cost. Training on massive datasets drains resources and introduces noise, creating a growing need for efficient and robust data selection (Wang et al., 2018; Csiba & Richtárik, 2018; Zheng et al., 2022; Katharopoulos & Fleuret, 2018). In practice, acquiring or maintaining such large datasets is often infeasible due to storage limits, privacy constraints, or annotation costs (Ganguli et al., 2022; Yang & Su, 2024). Coreset selection seeks to address this challenge by identifying a small, informative subset that preserves the performance of training on the full dataset (?Sorscher et al., 2022; Guo et al., 2022; Bhalerao, 2024). Beyond efficiency, coresets can improve robustness by excluding noisy, redundant, or overly difficult examples, reducing overfitting and sharpening generalization (Bengio et al., 2019; Katharopoulos & Fleuret, 2018). As highlighted by Sorscher et al. (2022), with the right pruning ratio, a well-selected coreset can even outperform full-data training, a surprising and powerful result, which has been verified in some applications (Na et al., 2021; Moser et al., 2022; Yao et al., 2023; Moser et al., 2024; Ding et al., 2023).

Despite these benefits, selecting an optimal coreset remains nontrivial (Zheng et al., 2022; Sener & Savarese, 2017). Most methods rely on gradient heuristics (Paul et al., 2021; Mirzasoleiman et al., 2020; Killamsetty et al., 2021a), influence estimation (Toneva et al., 2018; Paul et al., 2021), or decision boundary estimates (Ducoffe & Precioso, 2018; Margatina et al., 2021). Yet, current methods struggle with noise, computational overhead, and lack of class-awareness - key challenges in real-world applications (Zhang et al., 2021). Crucially, these approaches often prune via fixed sampling budgets rather than adapting to the natural density or ambiguity within each class (Agarwal et al., 2005; Sorscher et al., 2022; Guo et al., 2022; Zheng et al., 2022).

Our method, **HyperCore**, offers a new perspective. We train lightweight hypersphere models (Tax & Duin, 2004; Ruff et al., 2018; Liznerski et al., 2020) that learn to separate in-class from out-of-class samples in a class-conditional embedding space. Here, "out-of-class" refers not only to samples from other classes but also to mislabeled, ambiguous, or corrupted inputs that appear atypical when measured against a given class distribution. Treating such points as outliers is natural in a per-class setting, since they provide conflicting training signals for that class. By measuring the distance of each point to its class-specific hypersphere center, we obtain an interpretable conformity score.

To determine the hypersphere decision boundary, we adaptively select pruning thresholds without tuning hyperparameters. More explicitly, we exploit *Youden's J statistic* (Youden, 1950), a well-known criterion from signal detection theory, to filter uncertain or atypical examples in a per-class, data-driven way. This thresholding adapts automatically to class imbalance, ambiguity, and noise, in contrast to fixed global ratios.

HyperCore is computationally lightweight: each class-specific model is trained independently, enabling parallelization across classes; thresholding reduces to a one-pass scan over sorted distances, i.e., $O(n_c \log n_c)$ per class. In practice, this cost scales linearly with the number of classes and can be amortized by parallel workers or shared backbones.

Our contributions can be summarized as follows:

- We introduce a simple *class-wise hypersphere formulation* with fixed centers and pseudo-Huber loss, yielding interpretable conformity scores and avoiding costly center estimation.
- We propose *adaptive pruning via Youden's J statistic*, eliminating the need for global ratio tuning and naturally adjusting to per-class density and noise.
- We demonstrate *robustness under label noise and high pruning ratios*, where HyperCore outperforms state-of-the-art coreset selection methods across ImageNet-1K and CIFAR-10.
- We provide *scalability analysis*, showing that HyperCore remains efficient due to embarrassingly parallel training and near-linear complexity.

## 2 PRELIMINARIES

### 2.1 CORESET SELECTION

Consider a supervised learning setup, where the training set $\mathcal{T} = \{(\mathbf{x}_i, y_i)\}_{i=1}^{N}$ contains $N$ i.i.d. samples drawn from an unknown distribution $P$. Each input $\mathbf{x}_i \in \mathcal{X}$ is paired with a label $y_i \in \mathcal{Y}$.

**Definition 1** (Coreset Selection). *The goal is to extract a subset $\mathcal{S} \subset \mathcal{T}$ with $|\mathcal{S}| \ll |\mathcal{T}|$, such that training a model $\theta^{\mathcal{S}}$ on $\mathcal{S}$ achieves comparable generalization to training $\theta^{\mathcal{T}}$ on the full dataset $\mathcal{T}$:*

$$\mathcal{S}^* = \underset{\mathcal{S} \subset \mathcal{T}: \frac{|\mathcal{S}|}{|\mathcal{T}|} \approx 1-\alpha}{\arg\min} \mathbb{E}_{(\mathbf{x},y) \sim P} \left[ \mathcal{L}(\mathbf{x}, y; \theta^{\mathcal{S}}) - \mathcal{L}(\mathbf{x}, y; \theta^{\mathcal{T}}) \right], \quad (1)$$

*where $\alpha \in (0, 1)$ denotes the pruning ratio and $(1 - \alpha)$ is the retained fraction. $\mathcal{L}$ is the task-specific loss.*

**Notation.** Throughout the paper we use $\alpha$ for the pruning ratio and $(1 - \alpha)$ for the retained fraction.

While simple in form, this objective is difficult to achieve in practice. Effective coreset selection depends on identifying training samples that best support generalization. Popular approaches estimate sample importance via gradients, influence scores, or diversity heuristics (Nogueira et al., 2018; Song et al., 2022; Xiao et al., 2025; Zheng et al., 2022). Yet, these methods are often sensitive to noise, expensive to compute, or agnostic to class structure.

### 2.2 HYPERSPHERE CLASSIFIER

Hypersphere classifiers, such as Deep SVDD (Ruff et al., 2018) and FCDD (Liznerski et al., 2020), represent a class of anomaly detection methods that embed nominal data into a compact region of the feature space while mapping anomalies away.

**Definition 2** (Hypersphere Classifier). *Given a collection of samples $\mathbf{x}_1, \ldots, \mathbf{x}_n$ with labels $y_i \in \{0, 1\}$ (where $y_i = 0$ denotes a nominal sample and $y_i = 1$ denotes an anomaly), a hypersphere classifier seeks to learn a neural network mapping $\phi(\mathbf{x}; W)$ with parameters $W$ and a randomly (non-trivial) center $\mathbf{c} \in \mathbb{R}^d$ by optimizing the objective:*

$$\min_{W} \frac{1}{n} \sum_{i=1}^{n} \left[ (1 - y_i) h\left(\|\phi(\mathbf{x}_i; W) - \mathbf{c}\|\right) - y_i \log\left(1 - \exp\left(-h\left(\|\phi(\mathbf{x}_i; W) - \mathbf{c}\|\right)\right)\right) \right], \quad (2)$$

*where $h : \mathbb{R} \to \mathbb{R}$ is the pseudo-Huber loss (Huber, 1992) defined as $h(a) = \sqrt{a^2 + 1} - 1$. This loss function robustly penalizes deviations, interpolating from a quadratic penalty for small distances to a linear penalty for larger deviations.*

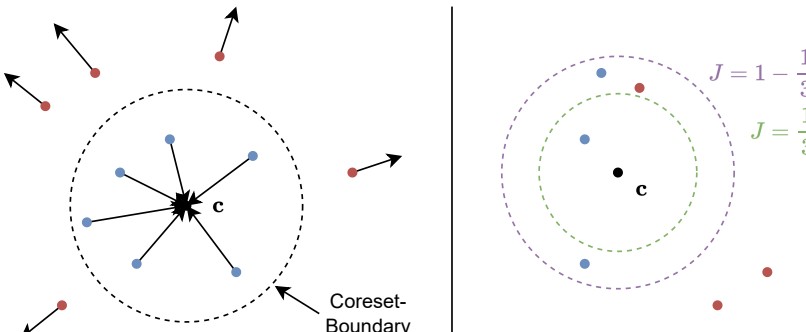

Figure 1: **Left:** Visualization of HyperCore. In-class samples are pulled toward the center, while out-of-class samples are pushed away, creating a clear separation. **Right:** Illustration of adaptive pruning ratio selection via Youden's J statistic. Two candidate thresholds are compared, with the purple threshold yielding a higher $J$ value and thus being considered more optimal for pruning.

**Coreset Selection Context.** This geometric intuition naturally facilitates threshold-based selection of representative data points, making hypersphere classifiers particularly suitable for robust coreset selection tasks. In a one-vs-rest view, we ask if a sample is *typical of class c*. Four sources tend to fall farther from the $c$-center in a class-conditional embedding: (i) true out-of-class samples, (ii) mislabeled instances whose content supports another class, (iii) borderline examples mixing evidence from multiple classes, and (iv) low-quality or corrupted inputs. All four inject conflicting gradients for class $c$, so excluding them when curating $c$'s coreset improves representativeness. In our experiments section, we show empirically that our derived coresets preferentially retain clean, central exemplars.

## 3 METHODOLOGY

Our method is built upon class-wise hypersphere models that assess the representativeness of each sample by measuring the distance to a fixed hypersphere center, as illustrated in Figure 1 (left). Specifically, we classify samples from the target class as normal, and all others as anomalies.

### 3.1 HYPERCORE LOSS WITH ZERO CENTER

A key simplification of HyperCore is anchoring the hypersphere center at the origin, i.e. $\mathbf{c} = \mathbf{0}$, which avoids explicit center estimation and enables lightweight optimization. Let $\phi(\mathbf{x}; W) \in \mathbb{R}^d$ denote the learned embedding of input $\mathbf{x}$ under parameters $W$. The embedding norm $\|\phi(\mathbf{x}; W)\|$ is then used as a measure of conformity.

**Definition 3** (HyperCore Loss). *Given a sample $\mathbf{x}$ with label $y \in \{0, 1\}$ (where $y = 0$ denotes an in-class sample and $y = 1$ an out-of-class sample), the HyperCore loss is defined as*

$$L_{HyperCore}(\mathbf{x}; W) = (1 - y)\, h(\|\phi(\mathbf{x}; W)\|) - y \, \log\left(1 - \exp\left(-h(\|\phi(\mathbf{x}; W)\|)\right)\right), \quad (3)$$

*where $h(a) = \sqrt{a^2 + 1} - 1$ is the pseudo-Huber loss (Huber, 1992). This loss penalizes small embedding norms for anomalies and encourages in-class samples to lie close to the origin.*

We now show that the trivial solution $W = 0$ (mapping all inputs to $\mathbf{0}$) is not optimal.

**Lemma 1** (Balanced Sampling Prevents Trivial Collapse). *Assume that each training batch is balanced, i.e. the number of in-class ($y = 0$) samples equals the number of out-of-class ($y = 1$) samples. Let $W_0$ be the all-zero weight configuration such that $\phi(\mathbf{x}; W_0) = \mathbf{0}$ for all $\mathbf{x}$. Then the HyperCore loss at $W_0$ is unbounded:*

$$L_{HyperCore}(\mathbf{x}; W_0) \to \infty, \quad (4)$$

*which rules out the trivial solution as optimal.*

*Proof.* At $W \to W_0$, we have $\phi(\mathbf{x}; W) \to \mathbf{0}$ and hence $\|\phi(\mathbf{x}; W)\| \to 0$. Since $h(0) = 0$, the loss for in-class samples ($y = 0$) vanishes. For out-of-class samples ($y = 1$), the term becomes

$$- \log\big(1 - \exp\big(-h(0)\big)\big) = -\log(1 - e^0) = -\log(0) \to \infty. \tag{5}$$

Thus, even a single out-of-class sample in a batch makes $L_{\text{HyperCore}}(\mathbf{x}; W_0)$ diverge. ∎

## 3.2 STATIC PRUNING: A BASELINE FOR COMPARISON

As a baseline, we adopt fixed pruning ratios. Let $\mathcal{T}_c^{\text{in}}$ denote the in-class dataset for class $c$. Each sample $\mathbf{x}_i \in \mathcal{T}_c^{\text{in}}$ is mapped to an embedding with norm

$$d_i = \|\phi_c(\mathbf{x}_i)\|. \tag{6}$$

Given a pruning fraction $\alpha$, we retain the $(1 - \alpha) \cdot |\mathcal{T}_c^{\text{in}}|$ samples with the smallest $d_i$. Formally,

$$\mathcal{S}_c^{\text{fixed}} = \{(\mathbf{x}_i, c) \in \mathcal{T}_c^{\text{in}} \mid d_i \leq \tau_c^{\text{fixed}}\}, \tag{7}$$

where $\tau_c^{\text{fixed}}$ is chosen such that exactly $(1 - \alpha) \cdot |\mathcal{T}_c^{\text{in}}|$ samples are kept. The global coreset is $\mathcal{S}^{\text{fixed}} = \bigcup_{c=0}^{C-1} \mathcal{S}_c^{\text{fixed}}$.

Although simple, this approach does not adapt to class-specific density or noise levels. Furthermore, finding an effective $\alpha$ typically requires testing multiple candidates, increasing overhead.

## 3.3 HYPERCORE WITH ADAPTIVE PRUNING RATIO

Instead of fixing $\alpha$, we determine class-specific thresholds via Youden's $J$ statistic (Youden, 1950). For each class $c$, let

$$D_c^{\text{in}} = \{d_i = \|\phi(\mathbf{x}_i; W)\| : (\mathbf{x}_i, c) \in \mathcal{T}_c^{\text{in}}\}, \text{ and}$$
$$D_c^{\text{out}} = \{d_j = \|\phi(\mathbf{x}_j; W)\| : (\mathbf{x}_j, c) \in \mathcal{T} \setminus \mathcal{T}_c^{\text{in}}\}. \tag{8}$$

For any candidate threshold $\tau$, we define

$$\text{TPR}_c(\tau) = \frac{|\{d \in D_c^{\text{in}} : d \leq \tau\}|}{|D_c^{\text{in}}|}, \qquad \text{FPR}_c(\tau) = \frac{|\{d \in D_c^{\text{out}} : d \leq \tau\}|}{|D_c^{\text{out}}|}. \tag{9}$$

Youden's $J$ statistic, as illustrated in Figure 1 (right), is then

$$J_c(\tau) = \text{TPR}_c(\tau) - \text{FPR}_c(\tau). \tag{10}$$

The optimal threshold is

$$\tau_c^* = \arg\max_{\tau \in D_c^{\text{in}}} J_c(\tau). \tag{11}$$

The class-specific coreset is

$$\mathcal{S}_c = \{(\mathbf{x}_i, c) \in \mathcal{T}_c^{\text{in}} \mid d_i \leq \tau_c^*\}, \tag{12}$$

and the global coreset is again the union $\mathcal{S} = \bigcup_{c=0}^{C-1} \mathcal{S}_c$.

**Lemma 2** (Threshold search complexity). *For class $c$, computing $\tau_c^* = \arg\max_{\tau \in D_c^{\text{in}}} J_c(\tau)$ requires sorting $D_c^{\text{in}}$ and a single linear scan, i.e., $O(n_c \log n_c)$ time and $O(n_c)$ memory. Summed over classes, the total time is $\sum_c O(n_c \log n_c)$ and memory $\sum_c O(n_c)$, with $\sum_c n_c = N$.*

## 3.4 PRACTICAL REMARKS AND COMPLEXITY

**Training overhead.** Each $\phi_c$ is a small network trained on $\mathcal{T}_c^{\text{in}} \cup \mathcal{T}_c^{\text{out}}$, typically much cheaper than full-dataset gradient-based selection.

**Thresholding overhead.** For each class, sorting $D_c^{\text{in}}$ costs $O(n_c \log n_c)$, where $n_c = |\mathcal{T}_c^{\text{in}}|$. Across classes, the complexity is near-linear in $N$.

**No fraction tuning.** HyperCore automatically derives class-specific pruning ratios without requiring $\alpha$. A global budget can be enforced if needed, but allowing classes to self-threshold often yields greater robustness.

Table 1: **Coreset selection performance on ImageNet-1K**. We evaluate various pruning methods by training randomly initialized ResNet-18 models on their selected subsets and testing on the full ImageNet validation set. DeepFool and GraNd are excluded due to their substantial memory and computational demands.

| Fraction $(1-\alpha)$ | 10% | 20% | 30% | 40% | 50% | 100% |
|---|---|---|---|---|---|---|
| Herding (Welling, 2009) | 29.17±0.23 | 41.26±0.43 | 48.71±0.23 | 54.65 ± 0.07 | 58.92 ± 0.19 | 69.52±0.45 |
| k-Center Greedy (Sener & Savarese, 2017) | 48.11±0.29 | 59.06±0.22 | 62.91±0.22 | 64.93 ± 0.22 | 66.04 ± 0.05 | 69.52±0.45 |
| Forgetting (Toneva et al., 2018) | **55.31±0.07** | **60.36±0.12** | 62.45±0.11 | 63.97 ± 0.01 | 65.06 ± 0.02 | 69.52±0.45 |
| CAL(Margatina et al., 2021) | 46.08±0.10 | 53.71±0.19 | 58.11±0.13 | 61.17±0.06 | 63.67 ±0.28 | 69.52±0.45 |
| Craig (Mirzasoleiman et al., 2020) | 51.39±0.13 | 59.33±0.22 | 62.72±0.13 | **64.96±0.00** | **66.29 ±0.00** | 69.52±0.45 |
| GradMatch (Killamsetty et al., 2021a) | 47.57±0.32 | 56.29±0.31 | 60.62±0.28 | 64.40±0.33 | 65.02 ± 0.50 | 69.52±0.45 |
| Glister (Killamsetty et al., 2021b) | 47.02±0.29 | 55.93±0.17 | 60.38±0.17 | 62.86±0.07 | 65.07±0.08 | 69.52±0.45 |
| **HyperCore (ours)** | 49.94±0.02 | 58.12±0.11 | **62.96±0.01** | **64.96±0.04** | 65.32±0.13 | 69.52±0.45 |

Table 2: **Fixed coreset selection accuracy on CIFAR-10** using randomly initialized ResNet-18 models (He et al., 2016). Bold entries indicate the highest performance at each data fraction.

| Fraction $(1-\alpha)$ | 0.1% | 0.5% | 1% | 5% | 10% | 20% | 30% | 40% | 50% | 60% | 90% | 100% |
|---|---|---|---|---|---|---|---|---|---|---|---|---|
| Herding (Welling, 2009) | 19.8±2.7 | 29.2±2.4 | 31.1±2.9 | 50.7±1.6 | 63.1±3.4 | 75.2±1.0 | 80.8±1.5 | 85.4±1.2 | 88.4±0.6 | 90.9±0.4 | 94.4±0.1 | 95.6±0.1 |
| k-Center Greedy (Sener & Savarese, 2017) | 19.9±0.9 | 25.3±0.9 | 32.6±1.6 | 55.6±2.8 | 74.6±0.9 | **87.3±0.2** | 91.0±0.3 | 92.6±0.2 | 93.5±0.5 | 94.3±0.2 | 95.5±0.1 | 95.6±0.1 |
| Forgetting (Toneva et al., 2018) | 21.3±1.2 | 29.7±0.3 | 35.6±1.0 | 51.1±2.0 | 66.9±2.0 | 86.6±1.0 | **91.7±0.3** | **93.0±0.2** | 94.1±0.2 | 94.6±0.2 | 95.4±0.1 | 95.6±0.1 |
| GraNd (Paul et al., 2021) | 14.6±0.8 | 17.2±0.8 | 18.6±0.8 | 28.9±0.5 | 41.3±1.3 | 71.1±1.3 | 88.3±1.0 | **93.0±0.4** | **94.8±0.1** | **95.2±0.1** | 95.5±0.1 | 95.6±0.1 |
| CAL (Margatina et al., 2021) | 23.1±1.8 | 31.7±0.9 | 39.7±3.8 | **60.8±1.4** | 69.7±0.8 | 79.4±0.9 | 85.1±0.7 | 87.6±0.3 | 89.6±0.4 | 90.9±0.4 | 94.7±0.2 | 95.6±0.1 |
| DeepFool (Ducoffe & Precioso, 2018) | 18.7±0.9 | 26.4±1.1 | 28.3±0.6 | 47.7±3.5 | 61.2±2.8 | 82.7±0.5 | 90.8±0.5 | 92.9±0.2 | 94.4±0.1 | 94.8±0.1 | **95.6±0.1** | 95.6±0.1 |
| Craig (Mirzasoleiman et al., 2020) | 19.3±0.3 | 29.1±1.6 | 32.8±1.8 | 42.5±1.7 | 59.9±2.1 | 78.1±2.5 | 90.0±0.5 | 92.8±0.2 | 94.3±0.2 | 94.8±0.1 | 95.5±0.1 | 95.6±0.1 |
| GradMatch (Killamsetty et al., 2021a) | 17.4±1.6 | 27.1±1.1 | 27.7±2.0 | 41.8±2.4 | 55.5±2.3 | 78.1±2.0 | 89.6±0.7 | 92.7±0.5 | 94.1±0.2 | 94.7±0.3 | 95.4±0.1 | 95.6±0.1 |
| Glister (Killamsetty et al., 2021b) | 18.4±1.3 | 26.5±0.7 | 29.4±1.9 | 42.1±1.0 | 56.8±1.8 | 77.2±2.4 | 88.8±0.6 | 92.7±0.4 | 94.2±0.1 | 94.8±0.2 | 95.5±0.1 | 95.6±0.1 |
| **HyperCore (ours)** | **24.5±1.3** | **35.4±1.0** | **40.7±1.0** | 60.3±1.3 | **71.1±0.9** | 83.5±0.5 | 88.6±0.4 | 91.1±0.3 | 92.3±0.1 | 93.1±0.1 | 95.0±0.2 | 95.6±0.1 |

## 4 EXPERIMENTS

In this section, we present experiments on ImageNet-1K (Deng et al., 2009) and CIFAR-10 (Krizhevsky et al., 2009) that assess HyperCore across several dimensions, including overall coreset quality, runtime efficiency, and robustness.

**Backbone and training.** For all experiments, we use ResNet-18 (He et al., 2016), following training protocols from DeepCore (Guo et al., 2022). Models are trained with SGD for 200 epochs using a cosine-annealed learning rate (initial 0.1), momentum 0.9, weight decay $5 \times 10^{-4}$, and standard data augmentation (random crop + flip). On CIFAR-10, we use a batch size of 128; on ImageNet-1K, a batch size of 256. To establish upper-bound references, we also train ResNet-18 on random subsets of varying fractions of the full dataset.

**HyperCore training.** Each class-specific HyperCore model is trained on balanced batches (half in-class, half out-of-class). We use Adam with learning rate $10^{-4}$, batch size 128 for CIFAR-10 and 512 for ImageNet, and train for 100 epochs per class. Since classes are independent, training runs fully in parallel, making the overall wall-clock cost scale with available compute rather than the number of classes.

**Robustness protocol.** To evaluate label noise tolerance, we adopt the poisoning setup of Zhang et al. (2021), injecting label noise and malicious relabeling into the training data.

### 4.1 IMAGENET-1K RESULTS

Our evaluation on ImageNet-1K in Table 1 demonstrates that HyperCore achieves consistently strong performance, positioning itself among the top-performing coreset selection methods despite being explicitly designed with robustness as its primary objective. Specifically, at moderate to high retained fractions (30–50%), HyperCore matches or slightly surpasses established state-of-the-art methods such as Craig and GradMatch. Even at more aggressive pruning (e.g., $(1-\alpha) = 10\%$–$20\%$ retained), HyperCore closely follows the best-performing methods and achieves highly competitive results.

Table 3: **Fixed coreset selection performance under label noise on CIFAR-10**, where 10% of the training labels are randomly corrupted by assigning them to incorrect classes. Bold entries indicate the highest performance at each data fraction.

| Fraction $(1-\alpha)$ | 0.1% | 0.5% | 1% | 5% | 10% | 20% | 30% | 40% | 50% | 60% | 90% | 100% |
|---|---|---|---|---|---|---|---|---|---|---|---|---|
| Herding (Welling, 2009) | 11.4±0.9 -8.4 | 10.8±0.5 -18.4 | 11.1±0.9 -20.0 | 10.6±1.1 -40.1 | 11.7±0.9 -51.4 | 26.0±3.4 -49.2 | 50.1±1.3 -30.7 | 71.0±0.6 -14.4 | 79.1±1.8 -9.3 | 84.6±0.6 -6.3 | 90.4±0.1 -4.0 | 90.8±0.1 -4.8 |
| k-Center Greedy (Sener & Savarese, 2017) | 12.6±1.3 -7.3 | 14.3±0.8 -11.0 | 16.1±1.0 -16.5 | 29.6±1.6 -26.0 | 41.7±3.0 -32.9 | 62.0±2.0 -25.3 | 73.8±1.8 -17.2 | 80.2±0.7 -12.4 | 83.9±0.7 -9.6 | 86.6±0.5 -7.7 | 90.4±0.3 -5.1 | 90.8±0.1 -4.8 |
| Forgetting (Toneva et al., 2018) | **21.8±1.6** +0.5 | 31.1±1.0 +1.4 | 35.0±1.3 -0.6 | 52.8±1.2 +1.7 | 66.4±1.3 -0.5 | 83.2±1.0 -3.4 | 88.9±0.2 -2.8 | 90.7±0.2 -2.3 | 91.0±0.4 -3.1 | 91.5±0.4 -3.1 | 90.9±0.1 -4.5 | 90.8±0.1 -4.8 |
| GraNd (Paul et al., 2021) | 11.5±0.9 -3.1 | 11.9±0.8 -5.3 | 11.1±0.6 -7.5 | 10.8±1.1 -18.1 | 10.6±1.2 -30.7 | 25.4±0.9 -45.7 | 44.8±2.0 -43.5 | 67.2±2.6 -25.8 | 79.4±1.3 -15.4 | 86.3±0.3 -8.9 | 90.2±0.2 -5.3 | 90.8±0.1 -4.8 |
| CAL (Margatina et al., 2021) | 21.3±1.7 -1.8 | 30.8±1.0 -0.9 | 36.8±1.3 -1.2 | 59.9±0.8 -0.9 | 71.3±1.0 -1.6 | 80.0±0.2 +0.6 | 83.9±0.6 -1.2 | 87.1±0.3 -0.5 | 89.1±0.2 -0.3 | 90.6±0.2 -0.3 | 91.9±0.1 -2.8 | 90.8±0.1 -4.8 |
| DeepFool (Ducoffe & Precioso, 2018) | 17.4±0.9 -1.3 | 21.6±1.4 -4.8 | 25.3±1.3 -3.0 | 33.6±0.4 -14.1 | 43.9±3.2 -17.3 | 65.5±1.6 -17.2 | 77.0±1.3 -13.8 | 84.5±0.5 -8.4 | 86.6±0.9 -7.8 | 88.7±0.5 -6.1 | 90.8±0.1 -4.8 | 90.8±0.1 -4.8 |
| Craig (Mirzasoleiman et al., 2020) | 19.5±1.4 +0.2 | 20.2±1.4 -8.9 | 24.8±1.1 -8.0 | 30.4±0.9 -12.1 | 31.7±1.6 -28.2 | 39.2±1.5 -38.9 | 58.4±2.9 -31.6 | 73.1±1.4 -19.7 | 81.4±0.6 -12.9 | 85.3±0.4 -9.5 | 90.5±0.3 -5.0 | 90.8±0.1 -4.8 |
| GradMatch (Killamsetty et al., 2021a) | 15.7±2.0 -1.7 | 21.4±0.7 -5.7 | 23.0±1.6 -4.7 | 27.6±2.4 -14.2 | 31.4±2.5 -24.1 | 37.7±2.1 -40.4 | 55.6±2.5 -34.0 | 72.0±1.4 -20.7 | 80.3±0.4 -13.8 | 85.3±0.5 -9.4 | 90.2±0.2 -5.2 | 90.8±0.1 -4.8 |
| Glister (Killamsetty et al., 2021b) | 14.9±2.0 -3.5 | 20.9±1.5 -5.6 | 24.5±1.3 -4.9 | 29.0±1.9 -13.1 | 31.7±2.1 -25.1 | 40.2±2.3 -37.0 | 57.2±1.3 -31.6 | 72.2±1.3 -20.5 | 80.6±0.4 -13.6 | 85.4±0.5 -9.4 | 90.2±0.2 -5.3 | 90.8±0.1 -4.8 |
| **HyperCore (ours)** | 20.9±1.5 -3.6 | **32.6±1.1** -2.8 | **41.4±1.0** +0.7 | **60.5±1.4** +0.2 | **70.0±0.9** -1.1 | **84.2±0.9** +0.7 | **89.1±0.4** +0.5 | **91.0±0.5** -0.1 | **92.6±0.1** +0.3 | **93.5±0.2** +0.4 | **93.7±0.3** -1.3 | 90.8±0.1 -4.8 |

## 4.2 CIFAR-10 Results

Comparisons to other coreset baselines are shown in Table 2. HyperCore achieves *up to 5.6% higher accuracy than the best baseline at aggressive pruning levels*. More specifically, HyperCore is on par for larger retained fractions while reaching the overall best results between $(1-\alpha) = 0.1\%$ and 10%. Furthermore, the parallelizable design of HyperCore enables us to obtain these outcomes with significantly reduced execution times, as shown in Figure 2.

In Table 3, we investigate how each coreset selection strategy holds up when 10% of the CIFAR-10 training labels are cor-

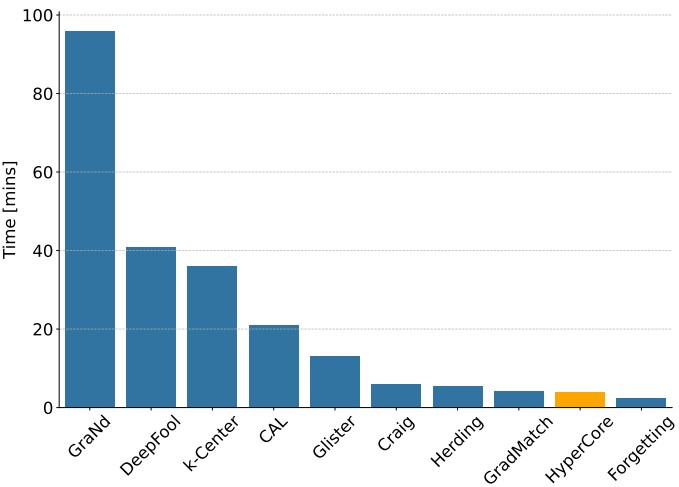

Figure 2: Time-Measurement on CIFAR-10. HyperCore ranks among the fastest techniques, including training, averaging only 4 minutes per class and benefiting from a parallelizable design.

rupted with random misassignments (Zhang et al., 2021). As one might expect, most approaches struggle at very small retained fractions (e.g., $(1-\alpha) = 0.1\%$ or 0.5%). In fact, if we look at the leftmost columns, methods such as GraNd or DeepFool perform particularly poorly, sometimes falling below 15% accuracy. Yet even at these high pruning levels, it is notable that Forgetting stands out with a slightly higher result at $(1-\alpha) = 0.1\%$.

For larger retained fractions (0.5% and beyond), one sees the advantages of HyperCore sharpen. For instance, at 1% of the data, HyperCore achieves 41.4% accuracy, surpassing the second-best method (CAL) by a margin of about 4.6%. This gap widens in the mid-range fractions (10%, 20%, 30%), underscoring the resilience of our method to label noise: whereas other approaches tend to plateau or fade noticeably, HyperCore keeps the performance or advances. As expected, accuracies converge at 90% dataset usage since almost all data is included. Notably, from 40% onward, HyperCore empirically validates the theory of Sorscher et al. (2022) by surpassing even full dataset performance. Further experiments (see appendix) with VGG-16, InceptionNet, ResNet-50, and WRN-16-8 confirm these observations: HyperCore is exceptionally efficient under noisy conditions.

### 4.3 ADAPTIVE CORESET SELECTION

The results presented in Table 4 demonstrate the effectiveness of adaptive coreset selection using HyperCore across various levels of label poisoning on CIFAR-10. When no label noise is present, HyperCore matches the accuracy obtained by training on the full dataset, illustrating that adaptive selection does not compromise model performance. More notably, as the level of label noise increases, HyperCore significantly outperforms training on the full dataset.

The Figure 3 shows that as the percentage of relabeling increases (i.e., as the level of label poisoning grows), the adaptive thresholds - the hypersphere radii - tend to increase. This behavior suggests that with more noise, the embeddings become more dispersed; in order to retain as many true inlier samples as possible, the model adapts by enlarging the decision boundary. Moreover, the rising standard deviation across classes indicates that the impact of label noise is not uniform: some classes experience a greater shift in their radii than others. In short, the model compensates for increased uncertainty by raising the threshold, which, though it might seem counterintuitive at first, is necessary to maintain robust discrimination between inliers and outliers under noisy conditions.

Table 4: CIFAR-10 performance for training on full dataset vs on adaptively derived coresets (HyperCore).

| Poisoning | Accuracy [%] | | $\alpha_{\text{HyperCore}}$ [%] |
|---|---|---|---|
| | Full Dataset | HyperCore | |
| 0% | **95.6±0.1** | **95.6±0.2** | 00.6±0.4 |
| 10% | 90.8±0.1 | **94.8±0.2** | 16.4±1.3 |
| 20% | 87.5±0.4 | **93.5±0.3** | 29.4±2.4 |
| 30% | 85.9±0.3 | **91.1±0.3** | 41.2±2.7 |
| 40% | 83.7±0.5 | **86.9±1.3** | 50.5±4.0 |

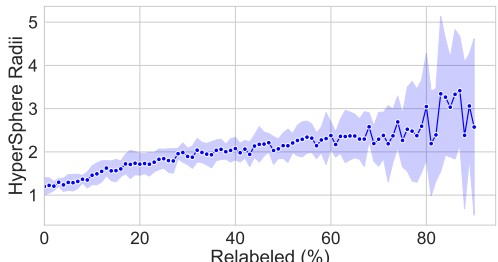

Figure 3: Average hypersphere radii (adaptive thresholds) and their standard deviations as a function of the relabeling percentage. The plot reveals that both the mean radius and its variability increase with higher levels of label poisoning, reflecting a broader dispersion in the embedding space and an adaptive expansion of the decision boundary to accommodate noise.

### 4.4 ANALYSIS OF YOUDEN'S J STATISTICS

Regarding adaptiveness, Figure 4 compares the key performance curves of **HyperCore** under increasing levels of artificially introduced label noise. In the left panel, we plot the confusion-based rates, namely True Positive Rate (**TPR**), False Positive Rate (**FPR**), True Negative Rate (**TNR**), and False Negative Rate (**FNR**), as a function of the poisoning percentage. Broadly speaking, positive rates denote included samples, while negative rates mean they are excluded. Despite the growing noise, we observe that TPR and TNR remain consistently higher than FPR and FNR, indicating that our hypersphere models selectively exclude corrupted or ambiguous samples. Meanwhile, FPR and FNR only moderately increase, suggesting that HyperCore successfully mitigates the risk of discarding genuine samples or retaining mislabeled ones.

In the right panel, we plot Youden's $J$ in orange, alongside the fraction of removed data in blue. Even as the fraction removed escalates for severe noise, Youden's $J$ remains relatively stable. This interplay demonstrates that the pruning decisions of HyperCore are not overly conservative. Although HyperCore discards an increasingly large portion of the data under extreme mislabeling, it still identifies informative inliers with sufficient reliability to maintain a viable Youden's $J$. Overall, the figure underscores strong resilience to label noise.

## 5 RELATED WORK

**Distance-Based Pruning and Anomaly Detection.** In parallel, a rich literature on anomaly or outlier detection capitalizes on distance metrics. One-class methods such as *Support Vector Data Description* (SVDD) (Tax & Duin, 2004) enclose normal samples in a minimal-radius hypersphere, labeling points outside as outliers. Deep SVDD extends this idea to a learned representation, forcing inliers near a randomly-sampled center in feature space (Ruff et al., 2018; Liznerski et al., 2020). These methods align with the *hypersphere* concept in **HyperCore**: Like one-class approaches, HyperCore identifies "inlier" samples with small norm while pushing outliers away. Unlike hypersphere

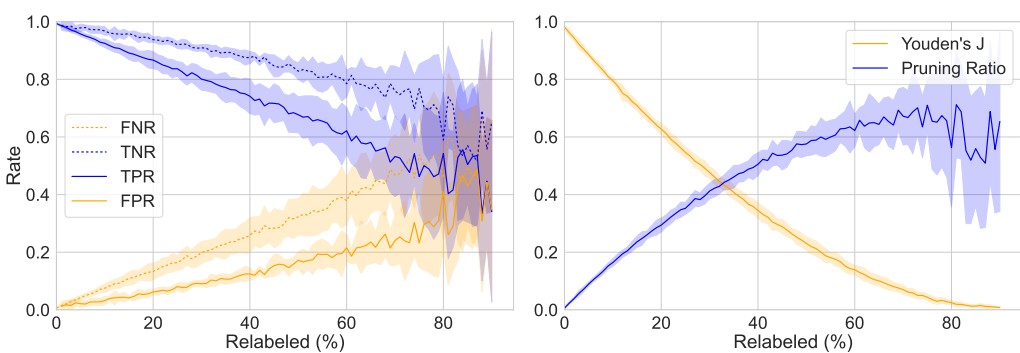

Figure 4: **Left:** Confusion-based metrics (TPR, FPR, TNR, FNR) under increasing label poisoning in CIFAR-10. **Right:** Youden's $J$ (orange) and fraction of removed samples (blue). Both plots highlight HyperCore's robust coreset selection behavior across varying degrees of poisoned labels (error-bands highlight the variance between the class labels).

classifiers, HyperCore uses simpler per-class MLPs with Youden's J thresholds, and a modified loss function.

**Norm-Based Confidence Scores in Distillation and Noise Removal.** Several approaches for dataset distillation or data pruning measure "confidence" using norms in feature space. For example, Lee et al. (Lee et al., 2018) compute class-conditional Gaussians on deep embeddings (Mahalanobis distance) and exclude points far from the nearest class center. Similarly, Pleiss et al. (Pleiss et al., 2020) track margin-based criteria to detect possible mislabels, effectively removing outlier examples. In coreset selection under noisy labels, Kang et al. (Kang et al., 2019) highlight that samples near the class centroid are typically correct, while label errors lie on the distribution fringe. This principle resonates with HyperCore's geometry-driven approach: we train a *binary in/out* classifier per class to separate inlier vs. out-class data, then threshold based on distance.

**Lightweight vs. Full-Model Coreset Approaches.** While gradient or influence-based selection can yield high-quality coresets (Paul et al., 2021; Killamsetty et al., 2021a), such methods typically incur substantial overhead: they require partial or entire model training to compute per-sample gradients or forgetting events (Toneva et al., 2018). By contrast, HyperCore remains *partially trained* (it fits a class-wise MLP to discriminate in/out), but each MLP sees only a subset of the dataset and does not require a large architecture or global alignment. This design reduces computation while flexibly adapting to each class's unique geometry.

**Prototype Selection and Continual Learning.** Prototype-based selection methods identify exemplars that approximate the class mean. For instance, iCaRL (Rebuffi et al., 2017) selects a small set of class representatives that minimize the distance to that class's mean embedding. When data are mislabeled or heavily imbalanced, however, simple centroid-based picks can inadvertently keep outliers if they exhibit subtle bias in embedding space. HyperCore addresses such issues by *actively* learning a boundary between inliers and outliers for each class, producing a more robust subset. In continual learning, HyperCore could replace iCaRL's herding by selecting reliably central class samples.

**Relation to Our HyperCore Approach.** We draw on the success of minimal, class-wise boundaries (like SVDD), but unify them with a simple *per-class in/out MLP* plus a *Youden's J threshold* to auto-prune ambiguous points. As a result, HyperCore effectively discards label noise and yields a representative coreset *without* requiring a large network or full-model backprop. This blend of geometry-driven inlier detection, threshold-based selection, and partial learning stands in contrast to existing coreset methods that rely on global fractions or heavy optimization of large models. HyperCore complements existing methods by providing a lightweight, noise-robust, class-specific alternative.

## 6 LIMITATIONS

HyperCore has several limitations that are worth noting despite its significant strengths. The hypersphere models depend heavily on learning meaningful embeddings from relatively small per-class data subsets. In classes with extremely limited or highly imbalanced data, the learned boundaries might degrade, reducing the robustness of HyperCore's adaptive pruning. Also, training separate hypersphere models per class could become computationally expensive as the number of classes scales (e.g., beyond thousands of classes) and the amount of GPUs/CPUs is limited, although HyperCore significantly reduces computational overhead compared to full-model coreset methods.

In addition, Youden's $J$ implicitly treats false positives and false negatives as equally costly. In domains with highly asymmetric costs or constraints (e.g., extreme class imbalance or safety-critical false negatives), alternative thresholding rules may be preferable—such as optimizing a weighted $J$ (cost-sensitive TPR/FPR), setting a target precision/recall operating point, or calibrating thresholds via validation risk minimization.

## 7 CONCLUSION & FUTURE WORK

We introduced HyperCore, a robust coreset selection framework leveraging hypersphere models. Unlike existing methods, HyperCore utilizes class-conditional embeddings with adaptive pruning thresholds determined by Youden's J statistic, enabling automatic and noise-aware subset selection without extensive hyperparameter tuning. HyperCore raises the bar for robust coreset selection, setting new benchmarks for pruning accuracy and label noise tolerance. By effectively discarding mislabeled or ambiguous data points, HyperCore ensures that the retained coresets are compact yet highly representative, thereby promoting efficient and robust model training.

Future work includes applying HyperCore to semi-supervised, continual learning, and large-scale tasks. Additionally, analyzing dynamic updates to hypersphere boundaries could further enhance HyperCore's versatility.

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
