# SUPPLEMENTARY MATERIAL FOR HYPERCORE: CORESET SELECTION UNDER NOISE VIA HYPERSPHERE MODELS

## A HARDWARE AND SOFTWARE

All experiments were run on a workstation equipped with an NVIDIA RTX A6000 GPU (48 GB VRAM). Our implementation uses PyTorch 1.10.1 with torchvision 0.11.2, and we build upon the DeepCore library for coreset selection routines.

## B DATASETS

We evaluated HyperCore on two widely used image-classification benchmarks:

- **CIFAR-10**: 50,000 training and 10,000 test images across 10 classes, each of size $32 \times 32$ pixels. Its small, balanced format makes it ideal for studying the behavior of coreset methods at extreme pruning levels.
- **ImageNet-1K**: 1,281,167 training and 50,000 validation images spanning 1,000 object categories. Following standard practice, we resize all images to $224 \times 224$ pixels before selection and training.

## C ADDITIONAL CROSS-ARCHITECTURE EVALUATIONS

Table 1: Coreset selection performances on CIFAR-10 with five randomly initialized VGG-16 models. The best results are marked in bold.

| Fraction $(1-\alpha)$ | 0.1% | 0.5% | 1% | 5% | 10% | 20% | 30% | 40% | 50% | 60% | 90% | 100% |
|---|---|---|---|---|---|---|---|---|---|---|---|---|
| Herding | 11.6±1.2 | 12.7±0.6 | 16.0±3.9 | 54.1±3.1 | 66.0±3.5 | 71.5±1.5 | 78.3±1.0 | 80.9±0.5 | 85.7±1.5 | 88.2±0.7 | 93.2±0.3 | 94.3±0.0 |
| k-Center Greedy | 12.3±1.0 | 14.5±2.9 | 14.7±1.3 | 38.7±12.0 | **75.6±1.3** | 85.5±0.3 | 89.0±0.4 | 91.0±0.2 | 91.8±0.3 | 92.7±0.2 | 94.1±0.1 | 94.3±0.0 |
| Forgetting | 12.5±0.8 | 13.8±2.8 | **25.6±4.3** | 55.9±1.3 | 75.2±1.4 | **86.2±0.2** | 89.0±0.2 | 91.3±0.2 | 91.9±0.2 | 92.8±0.0 | 94.1±0.2 | 94.3±0.0 |
| GraNd | 11.7±0.6 | 12.5±0.0 | 16.6±3.0 | 28.1±0.0 | 42.4±0.9 | 73.4±0.7 | 86.5±0.7 | 91.2±0.0 | 93.1±0.4 | **93.6±0.2** | 94.1±0.2 | 94.3±0.0 |
| CAL | 12.9±2.0 | 15.8±4.1 | 19.2±4.5 | **63.0±1.4** | 72.2±0.2 | 78.1±0.9 | 82.6±0.5 | 85.0±0.2 | 87.6±0.7 | 89.6±0.4 | 93.0±0.2 | 94.3±0.0 |
| DeepFool | 11.6±0.7 | 14.6±1.3 | 15.6±4.1 | 47.3±6.8 | 71.7±1.1 | 84.2±0.4 | **90.0±0.2** | **91.6±0.1** | 92.4±0.2 | 93.2±0.2 | 94.1±0.1 | 94.3±0.0 |
| Craig | 13.3±1.4 | 14.2±1.4 | 13.6±0.0 | 46.6±0.0 | 67.3±3.1 | 82.4±0.6 | 88.8±0.3 | 91.4±0.0 | 92.7±0.2 | 93.4±0.1 | **94.2±0.1** | 94.3±0.0 |
| GradMatch | 13.4±2.0 | 13.0±2.3 | 13.0±0.0 | 44.6±3.7 | 53.9±1.4 | 77.9±0.4 | 87.2±0.3 | 90.2±0.6 | **92.6±0.0** | 93.4±0.0 | **94.2±0.1** | 94.3±0.0 |
| Glister | **15.7±0.0** | 13.9±3.0 | 18.9±5.0 | 44.0±4.0 | 53.5±0.9 | 78.2±2.2 | 86.4±0.1 | 89.7±0.0 | 92.5±0.2 | 93.3±0.2 | **94.2±0.1** | 94.3±0.0 |
| **HyperCore (ours)** | **15.7±0.6** | **16.6±1.2** | 24.2±2.2 | 59.7±0.3 | 73.3±0.4 | 83.5±0.1 | 87.2±0.2 | 89.3±0.2 | 90.6±0.2 | 91.5±0.1 | 93.7±0.1 | 94.3±0.0 |

Table 2: Coreset selection performances on CIFAR-10 with five randomly initialized ResNet-50 models. The best results are marked in bold.

| Fraction $(1-\alpha)$ | 0.1% | 0.5% | 1% | 5% | 10% | 20% | 30% | 40% | 50% | 60% | 90% | 100% |
|---|---|---|---|---|---|---|---|---|---|---|---|---|
| Herding | 16.2±2.4 | 23.0±2.4 | 29.5±2.8 | 46.3±2.8 | 58.5±5.2 | 73.1±2.1 | 80.0±0.9 | 84.1±0.5 | 86.4±0.9 | 89.6±0.9 | 94.6±0.3 | 95.4±0.3 |
| k-Center Greedy | 15.4±1.8 | 18.8±3.1 | 25.0±4.6 | 53.1±1.9 | 68.8±2.1 | **86.6±0.4** | 90.2±0.6 | 91.7±0.2 | 93.4±0.3 | 94.2±0.3 | 95.3±0.3 | 95.4±0.3 |
| Forgetting | 17.0±2.2 | 26.9±2.7 | 34.0±1.0 | 49.5±2.9 | 67.5±2.6 | 86.2±0.7 | 89.9±0.4 | 92.0±0.3 | 92.8±0.2 | 93.6±0.1 | 95.2±0.1 | 95.4±0.3 |
| GraNd | 15.1±1.7 | 17.2±1.4 | 19.7±1.2 | 23.6±1.6 | 34.9±2.4 | 70.3±5.0 | 85.5±1.4 | 92.0±0.3 | **94.2±0.4** | **94.9±0.4** | **95.4±0.2** | 95.4±0.3 |
| CAL | 19.9±1.3 | **27.5±0.6** | 35.9±1.2 | **56.4±1.4** | 66.2±1.1 | 77.6±0.8 | 83.0±0.4 | 85.5±0.5 | 88.2±0.5 | 90.5±0.6 | 94.2±0.2 | 95.4±0.3 |
| DeepFool | 14.4±2.0 | 19.8±1.4 | 26.4±1.1 | 42.7±2.9 | 65.7±2.8 | 86.3±0.9 | **91.1±0.5** | **92.9±0.3** | 93.9±0.2 | 94.5±0.2 | 95.3±0.2 | 95.4±0.3 |
| Craig | 17.0±1.1 | 21.5±1.4 | 28.1±2.6 | 42.0±1.5 | 55.1±2.7 | 80.8±2.8 | 89.9±0.7 | 92.5±0.7 | 93.5±0.2 | 94.7±0.1 | **95.4±0.2** | 95.4±0.3 |
| GradMatch | 14.3±2.8 | 21.7±0.9 | 28.2±0.9 | 37.4±2.6 | 50.5±3.5 | 74.0±2.3 | 87.6±1.1 | 92.3±0.4 | 93.8±0.3 | 94.7±0.5 | 95.2±0.2 | 95.4±0.3 |
| Glister | 14.9±0.9 | 20.4±1.5 | 24.2±3.3 | 37.0±1.9 | 50.1±3.0 | 79.6±2.5 | 88.6±0.8 | 92.0±0.3 | 93.5±0.2 | 94.5±0.3 | 95.2±0.2 | 95.4±0.3 |
| **HyperCore (ours)** | **20.3±1.8** | 23.0±1.6 | **36.9±0.5** | **56.4±2.7** | **69.2±1.4** | 82.4±1.0 | 88.0±0.4 | 90.4±0.2 | 91.6±0.2 | 92.9±0.3 | 94.8±0.2 | 95.4±0.3 |

Table 3: Coreset selection performances on CIFAR-10 with five randomly initialized InceptionNetV3 models. The best results are marked in bold.

| Fraction $(1-\alpha)$ | 0.1% | 0.5% | 1% | 5% | 10% | 20% | 30% | 40% | 50% | 60% | 90% | 100% |
|---|---|---|---|---|---|---|---|---|---|---|---|---|
| Herding | 14.6±0.7 | 21.8±1.7 | 28.5±1.4 | 42.9±4.0 | 61.1±2.2 | 74.0±1.3 | 80.5±1.0 | 84.7±0.4 | 87.7±0.9 | 90.3±0.8 | 94.8±0.2 | 95.6±0.1 |
| k-Center Greedy | 15.1±2.0 | 22.1±3.0 | 26.6±1.4 | 51.2±2.3 | **72.9±1.7** | 85.8±0.4 | 89.8±0.4 | 92.2±0.3 | 93.6±0.3 | 94.4±0.4 | 95.4±0.2 | 95.6±0.1 |
| Forgetting | 18.5±0.6 | 28.5±1.0 | 32.2±0.9 | 50.9±1.5 | 69.3±1.3 | **85.4±0.9** | **90.4±0.4** | 92.4±0.1 | 93.4±0.2 | 94.3±0.3 | 95.4±0.2 | 95.6±0.1 |
| GraNd | 14.8±2.3 | 17.4±1.2 | 20.1±0.8 | 27.1±1.1 | 38.1±1.4 | 70.0±2.0 | 86.8±0.6 | **92.9±0.4** | **94.4±0.3** | **95.0±0.3** | 95.5±0.1 | 95.6±0.1 |
| CAL | 17.9±1.2 | 31.6±1.5 | 36.3±1.1 | **60.2±2.0** | 70.6±0.6 | 78.7±0.7 | 83.1±0.5 | 86.3±0.8 | 88.9±0.4 | 90.3±0.3 | 94.3±0.1 | 95.6±0.1 |
| DeepFool | 14.5±1.6 | 22.3±1.1 | 26.8±1.6 | 47.7±3.0 | 70.2±2.1 | 83.6±1.2 | 90.2±0.4 | 92.8±0.2 | 94.0±0.2 | 94.6±0.3 | **95.6±0.2** | 95.6±0.1 |
| Craig | 16.5±2.5 | 24.7±1.3 | 27.8±2.0 | 44.5±1.8 | 58.8±2.5 | 79.5±1.4 | 88.0±1.1 | 92.3±0.2 | 94.0±0.2 | **95.0±0.2** | **95.6±0.1** | 95.6±0.1 |
| GradMatch | 16.5±1.2 | 24.2±0.8 | 29.1±1.0 | 40.1±2.7 | 53.1±2.2 | 76.9±2.6 | 87.1±0.3 | 91.8±0.7 | 94.2±0.5 | 94.9±0.3 | 95.4±0.2 | 95.6±0.1 |
| Glister | 14.8±1.5 | 22.9±1.8 | 29.3±1.8 | 41.3±1.8 | 52.4±3.1 | 77.5±1.7 | 87.8±0.7 | 91.8±0.4 | 94.0±0.1 | 94.9±0.1 | **95.6±0.1** | 95.6±0.1 |
| **HyperCore (ours)** | **20.8±0.7** | **32.6±0.8** | **37.9±0.6** | 57.2±1.6 | 71.0±0.8 | 83.2±0.9 | 87.5±0.4 | 90.2±0.2 | 92.4±0.2 | 93.4±0.3 | 95.2±0.1 | 95.6±0.1 |

Table 4: Coreset selection performances on CIFAR-10 with five randomly initialized WRN-16-8 models. The best results are marked in bold.

| Fraction $(1-\alpha)$ | 0.1% | 0.5% | 1% | 5% | 10% | 20% | 30% | 40% | 50% | 60% | 90% | 100% |
|---|---|---|---|---|---|---|---|---|---|---|---|---|
| Herding | 23.8±1.4 | 31.3±3.5 | 39.6±2.8 | 60.2±1.2 | 66.9±2.3 | 77.3±1.0 | 81.9±0.7 | 86.4±1.3 | 89.0±0.6 | 92.3±1.0 | 95.2±0.1 | 96.0±0.1 |
| k-Center Greedy | 19.2±0.6 | 27.7±0.6 | 34.6±0.2 | 67.9±0.8 | **81.6±0.5** | **89.4±0.2** | 92.1±0.3 | 93.6±0.2 | 94.3±0.1 | 94.9±0.1 | 95.9±0.1 | 96.0±0.1 |
| Forgetting | 21.5±1.0 | 30.1±0.9 | 36.3±0.6 | 58.5±1.0 | 74.9±0.8 | 88.8±0.5 | **93.3±0.1** | **94.5±0.1** | 95.0±0.1 | 95.4±0.1 | 95.9±0.1 | 96.0±0.1 |
| GraNd | 15.0±1.8 | 19.5±0.7 | 21.4±0.3 | 37.8±1.0 | 58.7±1.0 | 81.8±0.7 | 92.1±0.1 | 94.3±0.2 | **95.3±0.1** | **95.7±0.2** | **96.0±0.2** | 96.0±0.1 |
| CAL | 22.2±2.4 | 37.1±2.5 | 45.5±1.4 | 66.2±0.6 | 74.2±0.9 | 82.7±0.5 | 86.3±0.8 | 89.2±0.6 | 91.0±0.3 | 92.4±0.2 | 95.3±0.1 | 96.0±0.1 |
| DeepFool | 18.9±1.9 | 29.2±1.0 | 35.0±1.8 | 57.4±3.2 | 74.0±1.8 | 87.5±0.4 | 92.0±0.3 | 93.5±0.2 | 94.6±0.1 | 95.2±0.1 | **96.0±0.1** | 96.0±0.1 |
| Craig | 21.9±1.5 | 30.0±0.8 | 38.1±1.4 | 60.1±1.4 | 69.2±0.8 | 86.5±0.4 | 91.4±0.2 | 93.8±0.1 | 94.8±0.1 | 95.4±0.1 | 95.9±0.1 | 96.0±0.1 |
| GradMatch | 20.1±1.8 | 27.8±1.2 | 31.4±2.1 | 54.4±2.6 | 70.5±2.0 | 84.6±0.7 | 90.7±0.5 | 93.3±0.3 | 94.7±0.4 | 95.3±0.1 | **96.0±0.1** | 96.0±0.1 |
| Glister | 20.1±1.2 | 28.0±1.4 | 31.6±0.8 | 49.3±3.3 | 67.9±1.7 | 83.6±0.8 | 90.5±0.9 | 93.6±0.1 | 94.6±0.1 | 95.2±0.2 | **96.0±0.2** | 96.0±0.1 |
| **HyperCore (ours)** | **26.9±0.7** | **39.3±0.2** | **45.9±0.2** | **67.4±0.3** | 79.1±0.3 | 86.5±0.2 | 90.3±0.1 | 92.2±0.2 | 93.2±0.2 | 94.0±0.2 | 95.6±0.1 | 96.0±0.1 |

Table 5: Fixed coreset selection performance of five randomly initialized VGG-16 under label noise on CIFAR-10, where 10% of the training labels are randomly corrupted by assigning them to incorrect classes. Bold entries indicate the highest performance at each data fraction.

| Fraction $(1-\alpha)$ | 0.1% | 0.5% | 1% | 5% | 10% | 20% | 30% | 40% | 50% | 60% | 90% | 100% |
|---|---|---|---|---|---|---|---|---|---|---|---|---|
| Herding | 12.5±0.7 (+0.9) | 11.7±1.6 (-1.0) | 11.8±1.4 (-4.2) | 10.5±0.8 (-43.6) | 14.6±2.8 (-51.4) | 33.7±4.3 (-37.9) | 57.2±2.0 (-21.1) | 71.9±1.8 (-9.0) | 81.1±0.7 (-4.6) | 83.3±0.4 (-5.0) | 88.6±0.3 (-4.5) | 88.8±0.2 (-5.5) |
| k-Center Greedy | 12.0±1.4 (-0.4) | 12.2±1.3 (-2.2) | 12.6±1.2 (-2.1) | 17.0±4.7 (-21.6) | 40.6±3.8 (-35.0) | 68.2±1.7 (-17.4) | 76.0±0.6 (-13.1) | 79.6±0.5 (-11.3) | 82.6±0.3 (-9.2) | 85.1±0.3 (-7.6) | 88.3±0.2 (-5.8) | 88.8±0.2 (-5.5) |
| Forgetting | 12.6±0.9 (+0.1) | 14.4±1.2 (+0.6) | 21.5±5.0 (-4.1) | 55.8±1.0 (-0.1) | **72.5±2.0** (-2.8) | 83.0±0.8 (-3.1) | 86.0±0.8 (-2.9) | 87.8±0.2 (-3.5) | 88.1±0.4 (-3.8) | 88.2±0.3 (-4.6) | 88.6±0.1 (-5.4) | 88.8±0.2 (-5.5) |
| GraNd | 12.8±1.2 (+1.1) | 11.4±0.5 (-1.1) | 12.1±1.1 (-4.5) | 10.9±0.4 (-17.3) | 11.1±0.4 (-31.3) | 24.9±0.8 (-48.5) | 45.4±0.6 (-41.1) | 64.1±2.2 (-27.1) | 76.2±0.9 (-17.0) | 82.9±0.6 (-10.8) | 88.8±0.2 (-5.3) | 88.8±0.2 (-5.5) |
| CAL | 13.9±1.5 (+0.9) | 13.2±1.3 (-2.6) | 18.0±5.3 (-1.2) | 57.6±2.0 (-5.4) | 69.6±2.1 (-2.6) | 76.1±0.7 (-1.9) | 81.8±0.5 (-0.8) | 84.1±0.6 (-0.9) | 86.3±0.4 (-1.3) | 88.0±0.3 (-1.6) | 89.6±0.4 (-3.4) | 88.8±0.2 (-5.5) |
| DeepFool | 13.1±2.0 (+1.5) | 13.2±1.0 (-1.4) | 17.4±2.2 (+1.8) | 36.1±4.3 (-11.2) | 62.5±2.9 (-9.2) | 74.5±0.7 (-9.6) | 82.0±0.4 (-8.0) | 84.8±0.3 (-6.8) | 86.5±0.4 (-6.0) | 87.1±0.2 (-6.0) | 88.6±0.1 (-5.5) | 88.8±0.2 (-5.5) |
| Craig | 13.7±2.0 (+0.3) | 11.5±0.3 (-2.7) | 13.0±1.5 | 22.8±2.1 (-23.8) | 36.0±3.3 (-31.3) | 55.2±4.6 (-27.1) | 67.3±1.8 (-21.4) | 78.5±1.5 (-12.9) | 82.2±0.5 (-10.5) | 85.3±0.4 (-8.1) | 88.7±0.3 (-5.5) | 88.8±0.2 (-5.5) |
| GradMatch | 12.1±0.8 (-1.3) | 12.7±1.4 (-0.4) | 13.7±2.9 (+0.7) | 31.6±1.9 (-13.0) | 41.5±2.6 (-12.4) | 48.3±2.3 (-29.6) | 61.4±1.4 (-25.9) | 72.7±1.1 (-17.5) | 79.0±1.5 (-13.6) | 84.6±0.4 (-8.8) | 88.8±0.3 (-5.5) | 88.8±0.2 (-5.5) |
| Glister | 12.7±1.4 (-2.9) | 13.0±0.7 (-0.9) | 13.7±2.4 (-5.2) | 30.4±1.0 (-13.6) | 42.6±3.2 (-11.0) | 49.3±3.8 (-28.9) | 61.5±1.0 (-24.9) | 71.1±2.3 (-18.6) | 79.3±1.5 (-13.2) | 84.5±0.4 (-8.8) | 88.7±0.3 (-5.5) | 88.8±0.2 (-5.5) |
| **HyperCore (ours)** | **14.0±0.6** (-1.8) | **21.9±3.4** (5.3) | **26.0±6.1** (1.7) | **62.6±1.9** (2.9) | 70.7±3.4 (-2.6) | **83.4±0.3** (-0.1) | **87.3±0.3** (0.0) | **89.5±0.3** (0.2) | **91.0±0.3** (0.4) | **92.1±0.1** (0.6) | **92.2±0.1** (-1.5) | 88.8±0.2 (-5.5) |

Table 6: Fixed coreset selection performance of five randomly initialized ResNet-50 under label noise on CIFAR-10, where 10% of the training labels are randomly corrupted by assigning them to incorrect classes. Bold entries indicate the highest performance at each data fraction.

| Fraction $(1-\alpha)$ | 0.1% | 0.5% | 1% | 5% | 10% | 20% | 30% | 40% | 50% | 60% | 90% | 100% |
|---|---|---|---|---|---|---|---|---|---|---|---|---|
| Herding | 11.4±1.1 (-4.8) | 12.4±1.9 (-10.6) | 10.6±0.7 (-18.9) | 11.4±0.8 (-34.9) | 11.3±0.7 (-47.2) | 36.5±3.8 (-36.7) | 56.6±2.1 (-23.4) | 71.5±2.6 (-12.6) | 79.3±1.0 (-7.1) | 84.5±0.8 (-5.1) | 89.7±0.3 (-4.9) | 89.8±0.4 (-5.6) |
| k-Center Greedy | 11.0±0.6 (-4.4) | 14.0±0.9 (-4.8) | 14.8±1.8 (-10.3) | 27.7±1.4 (-25.4) | 42.5±3.1 (-26.3) | 64.6±2.9 (-22.0) | 75.7±1.7 (-14.4) | 81.6±0.7 (-10.1) | 84.6±0.6 (-8.9) | 86.1±0.4 (-8.1) | 89.4±0.4 (-5.9) | 89.8±0.4 (-5.6) |
| Forgetting | 17.6±3.3 (+0.6) | 26.0±4.1 (-0.9) | 32.6±2.1 (-1.4) | 53.8±2.5 (+4.3) | **68.9±3.2** (+1.4) | 82.4±0.8 (-3.8) | 87.1±0.7 (-2.8) | 88.5±0.3 (-2.9) | 89.9±0.3 (-3.4) | 90.2±0.3 | 90.1±0.4 | 89.8±0.4 (-5.6) |
| GraNd | 13.5±1.3 (-1.6) | 13.5±0.8 (-3.6) | 12.8±1.7 (-6.9) | 11.3±1.1 (-12.3) | 12.0±1.2 (-22.9) | 22.4±1.3 (-48.0) | 44.8±1.5 (-40.7) | 62.9±1.4 (-29.2) | 77.1±0.2 (-17.1) | 84.6±1.0 (-10.3) | 89.5±0.4 (-6.0) | 89.8±0.4 (-5.6) |
| CAL | 16.9±4.1 (-3.0) | 27.0±2.2 (-2.5) | 34.5±2.4 (-2.4) | 52.2±2.9 (-4.1) | 65.2±0.3 (-1.0) | 76.8±1.2 (-0.8) | 81.7±1.0 (-1.3) | 84.5±0.5 (-1.1) | 86.9±0.4 (-1.3) | 88.5±0.5 (-2.0) | 90.5±0.1 (-3.7) | 89.8±0.4 (-5.6) |
| DeepFool | 16.0±3.4 (+1.6) | 19.8±1.2 (0.0) | 24.5±1.6 (-1.9) | 38.0±1.2 (-4.7) | 52.4±4.1 (-13.4) | 75.0±2.6 (-11.3) | 84.1±0.5 (-7.0) | 86.2±0.5 (-6.7) | 87.8±0.5 (-6.1) | 88.7±0.4 (-5.8) | 90.1±0.2 (-5.2) | 89.8±0.4 (-5.6) |
| Craig | 17.0±1.0 (-0.1) | 15.5±2.3 (-6.0) | 20.6±2.1 (-7.5) | 31.9±0.9 (-10.1) | 33.8±3.0 (-21.3) | 46.7±2.4 (-34.0) | 65.9±3.5 (-24.0) | 75.6±2.1 (-16.8) | 83.3±0.3 (-10.2) | 85.6±0.3 (-9.1) | 89.7±0.2 (-5.7) | 89.8±0.4 (-5.6) |
| GradMatch | 13.9±1.7 (-0.4) | 17.7±1.8 (-4.0) | 24.1±2.0 (-4.1) | 29.2±0.5 (-8.3) | 35.7±2.7 (-14.8) | 46.9±6.1 (-27.0) | 59.3±2.6 (-28.4) | 73.1±1.1 (-19.1) | 80.9±1.0 (-12.9) | 84.4±0.3 (-10.3) | 89.6±0.4 (-5.6) | 89.8±0.4 (-5.6) |
| Glister | 13.6±0.9 (-1.3) | 18.9±1.9 (-1.6) | 22.0±2.1 (-2.2) | 28.6±2.0 (-8.4) | 34.5±2.1 (-15.6) | 47.4±4.1 (-32.2) | 58.6±0.9 (-30.0) | 71.5±2.6 (-20.5) | 80.3±0.6 (-13.2) | 84.6±1.2 (-9.9) | 89.2±0.5 (-6.1) | 89.8±0.4 (-5.6) |
| **HyperCore (ours)** | **20.3±1.4** (-0.1) | **28.8±1.7** (-1.2) | **36.4±1.6** (-0.5) | **56.1±2.4** (-0.2) | 68.5±0.7 (-0.8) | **83.5±0.7** (+1.1) | **87.8±0.3** (-0.2) | **90.7±0.3** (+0.3) | **92.1±0.1** (+0.5) | **93.1±0.2** (+0.1) | **93.2±0.3** (-1.6) | 89.8±0.4 (-5.6) |

Table 7: Fixed coreset selection performance of five randomly initialized InceptionNetV3 under label noise on CIFAR-10, where 10% of the training labels are randomly corrupted by assigning them to incorrect classes. Bold entries indicate the highest performance at each data fraction.

| Fraction $(1-\alpha)$ | 0.1% | 0.5% | 1% | 5% | 10% | 20% | 30% | 40% | 50% | 60% | 90% | 100% |
|---|---|---|---|---|---|---|---|---|---|---|---|---|
| Herding | 12.4±1.2
-2.2 | 12.1±0.6
-9.7 | 10.8±0.7
-17.7 | 10.1±0.2
-32.9 | 10.6±0.6
-50.5 | 34.5±4.0
-39.5 | 54.5±2.3
-26.1 | 69.3±1.8
-15.4 | 77.9±1.0
-9.9 | 83.8±1.1
-6.5 | 90.9±0.4
-3.9 | 90.9±0.4
-4.6 |
| k-Center Greedy | 12.7±1.8
-2.4 | 13.6±0.8
-8.5 | 14.2±0.8
-12.4 | 27.6±1.9
-23.6 | 45.7±3.8
-27.1 | 66.1±1.4
-19.7 | 73.1±1.0
-16.6 | 81.0±0.5
-11.2 | 83.7±0.8
-9.9 | 86.4±0.7
-7.9 | 90.2±0.3
-5.2 | 90.9±0.4
-4.6 |
| Forgetting | 17.1±2.4
-1.4 | 28.6±0.7
0.0 | 33.1±1.7
+0.9 | 50.0±1.0
-1.0 | 71.2±0.8
+1.9 | 83.1±1.2
-2.3 | 87.3±0.7
-3.1 | 89.4±0.1
-2.9 | 90.2±0.3
-3.1 | 90.6±0.3
-3.7 | 90.8±0.3
-4.6 | 90.9±0.4
-4.6 |
| GraNd | 12.9±1.0
-1.9 | 12.5±1.0
-4.9 | 11.2±0.6
-8.9 | 10.9±0.5
-16.2 | 11.3±0.3
-26.8 | 25.0±1.0
-45.0 | 44.9±1.0
-41.9 | 61.8±1.9
-31.0 | 77.0±0.9
-17.4 | 83.3±0.9
-11.6 | 90.6±0.3
-5.0 | 90.9±0.4
-4.6 |
| CAL | 15.9±1.5
-2.1 | 26.8±1.9
-4.8 | 30.9±1.6
-5.4 | 56.3±3.5
-3.9 | 67.8±2.7
-2.7 | 77.0±0.7
-1.7 | 82.3±0.8
-0.7 | 84.5±0.5
-1.8 | 87.1±0.3
-1.8 | 89.1±0.3
-1.2 | 91.5±0.3
-2.8 | 90.9±0.4
-4.6 |
| DeepFool | 17.0±1.0
+2.4 | 19.9±1.5
-2.4 | 24.7±2.2
-2.1 | 38.1±1.0
-9.7 | 54.7±5.2
-15.5 | 76.1±0.7
-7.5 | 83.2±0.6
-7.0 | 87.2±0.5
-5.5 | 89.1±0.3
-5.0 | 89.9±0.4
-4.7 | 91.3±0.2
-4.2 | 90.9±0.4
-4.6 |
| Craig | 17.8±2.5
+1.3 | 21.0±1.1
-3.7 | 23.4±1.4
-0.4 | 29.8±2.0
-14.7 | 36.2±3.2
-22.6 | 45.0±3.5
-34.4 | 58.6±2.2
-29.4 | 71.4±1.1
-20.9 | 78.5±1.1
-15.5 | 84.3±0.8
-10.6 | 90.7±0.4
-4.9 | 90.9±0.4
-4.6 |
| GradMatch | 14.4±1.6
-2.1 | 21.5±1.0
-2.7 | 22.2±1.0
-6.9 | 29.3±2.4
-10.8 | 38.3±3.3
-14.7 | 42.7±3.3
-34.2 | 57.4±0.9
-29.7 | 68.3±0.8
-23.5 | 75.3±2.0
-19.0 | 83.1±0.8
-11.8 | 90.8±0.2
-5.0 | 90.9±0.4
-4.6 |
| Glister | 15.0±1.4
+0.2 | 20.2±1.4
-2.7 | 22.9±0.7
-6.3 | 32.3±1.7
-9.0 | 38.3±3.4
-14.0 | 43.1±2.3
-34.4 | 54.8±3.6
-33.0 | 67.4±2.9
-24.4 | 77.0±1.0
-17.1 | 82.9±0.6
-12.0 | 90.9±0.2
-4.7 | 90.9±0.4
-4.6 |
| **HyperCore (ours)** | **20.1±1.8**
-0.7 | **31.2±0.2**
-1.3 | **37.4±0.7**
-0.5 | **57.3±1.8**
0.0 | **72.8±1.6**
+1.8 | **83.5±0.3**
+0.2 | **87.6±0.3**
0.0 | **90.4±0.2**
+0.3 | **92.3±0.3**
-0.1 | **93.5±0.3**
+0.1 | **93.6±0.3**
-1.6 | 90.9±0.4
-4.6 |

Table 8: Fixed coreset selection performance of five randomly initialized WRN-16-8 under label noise on CIFAR-10, where 10% of the training labels are randomly corrupted by assigning them to incorrect classes. Bold entries indicate the highest performance at each data fraction.

| Fraction $(1-\alpha)$ | 0.1% | 0.5% | 1% | 5% | 10% | 20% | 30% | 40% | 50% | 60% | 90% | 100% |
|---|---|---|---|---|---|---|---|---|---|---|---|---|
| Herding | 11.3±0.8
-12.5 | 10.7±1.6
-20.5 | 8.9±1.7
-30.7 | 8.6±1.4
-51.6 | 12.9±2.5
-53.9 | 32.8±1.7
-44.5 | 56.0±1.5
-25.9 | 71.7±1.6
-14.7 | 79.5±1.0
-9.6 | 85.1±0.8
-7.2 | 91.3±0.3
-3.9 | 91.6±0.2
-4.4 |
| k-Center Greedy | 12.6±1.0
-6.6 | 15.0±0.9
-12.7 | 18.3±0.7
-16.3 | 31.4±1.6
-36.5 | 47.8±3.9
-33.8 | 68.8±2.3
-20.6 | 78.3±0.9
-13.8 | 82.4±0.6
-11.2 | 85.5±0.3
-8.8 | 87.8±0.2
-7.2 | 91.1±0.2
-4.8 | 91.6±0.2
-4.4 |
| Forgetting | 21.7±2.2
+0.2 | 31.7±1.6
+1.6 | 36.0±1.5
+3.5 | 62.0±0.7
+2.6 | 77.4±0.7
-1.7 | **88.1±0.2**
-2.4 | **91.0±0.1**
-3.0 | 91.5±0.3
-3.2 | 91.8±0.3
-3.7 | 91.7±0.2
-2.5 | 91.5±0.3
-2.6 | 91.6±0.2
-4.4 |
| GraNd | 12.7±1.7
-2.3 | 10.8±0.5
-8.8 | 8.7±0.9
-12.8 | 8.6±0.7
-29.1 | 10.5±0.8
-48.2 | 29.2±1.1
-52.5 | 50.5±1.2
-41.6 | 67.1±0.7
-27.2 | 79.5±0.6
-15.8 | 84.9±0.4
-10.7 | 91.4±0.2
-4.6 | 91.6±0.2
-4.4 |
| CAL | **24.9±2.0**
+2.6 | 36.2±1.3
-1.0 | 44.2±2.2
-1.3 | 67.4±1.3
+1.2 | 73.9±0.9
-0.2 | 82.5±0.5
-0.2 | 85.5±0.6
-0.7 | 88.7±0.3
-0.4 | 90.6±0.3
-0.4 | 91.6±0.3
-0.9 | 92.8±0.2
-2.6 | 91.6±0.2
-4.4 |
| DeepFool | 17.2±1.7
-1.7 | 22.2±1.4
-7.0 | 27.3±2.3
-7.7 | 43.2±1.7
-14.2 | 54.7±4.1
-19.4 | 70.4±1.7
-17.1 | 80.4±1.5
-11.6 | 84.7±0.6
-8.8 | 87.7±0.4
-6.9 | 89.3±0.2
-5.9 | 91.3±0.2
-4.7 | 91.6±0.2
-4.4 |
| Craig | 20.6±0.9
-1.3 | 24.6±2.3
-5.4 | 27.0±2.5
-11.1 | 28.5±1.8
-31.6 | 33.2±1.8
-36.0 | 46.8±4.0
-39.6 | 62.5±1.8
-28.8 | 74.6±0.6
-19.2 | 82.2±0.5
-12.5 | 86.5±1.0
-8.9 | 91.2±0.2
-4.7 | 91.6±0.2
-4.4 |
| GradMatch | 16.4±1.7
-3.8 | 23.6±1.8
-4.2 | 24.9±2.6
-6.6 | 29.9±2.6
-24.5 | 36.9±5.3
-33.5 | 49.7±3.7
-34.9 | 63.9±4.2
-26.8 | 75.7±2.0
-17.6 | 80.9±0.9
-13.8 | 86.1±0.3
-9.3 | 91.1±0.1
-4.8 | 91.6±0.2
-4.4 |
| Glister | 17.1±0.7
-3.0 | 24.8±1.4
-3.3 | 28.8±2.6
-2.8 | 28.3±2.9
-21.0 | 39.3±2.7
-28.5 | 47.8±3.8
-35.8 | 63.4±5.0
-27.1 | 74.0±1.2
-19.5 | 81.5±0.4
-13.2 | 86.5±0.2
-8.7 | 91.0±0.1
-4.9 | 91.6±0.2
-4.4 |
| **HyperCore (ours)** | 22.7±0.8
-4.3 | **39.6±0.8**
+0.4 | **47.4±0.5**
+1.5 | **70.2±0.7**
+2.8 | **79.2±0.5**
+0.1 | 86.7±0.2
+0.2 | 90.0±0.3
-0.2 | **92.2±0.1**
0.0 | **93.4±0.1**
+0.2 | **94.2±0.2**
+0.2 | 94.1±0.1
-1.5 | 91.6±0.2
-4.4 |