# OpenReview forum: "HyperCore: Coreset Selection under Noise via Hypersphere Models"
_ICLR.cc/2026/Conference — ICLR 2026 Conference Withdrawn Submission_

### Official Review · Reviewer_1PsK · 2025-10-28

**Soundness:** 1
**Presentation:** 3
**Contribution:** 4
**Rating:** 4
**Confidence:** 4

**Summary:**

This paper introduces HyperCore, a robust and adaptive coreset selection framework designed explicitly for noisy environments. Unlike existing methods, HyperCore utilizes class-conditional embeddings derived by hypersphere models with adaptive pruning thresholds determined by Youden's J statistic, enabling automatic and noise-aware subset selection without extensive hyperparameter tuning. Experiments on CIFAR-10 with 10% label noise show the effectiveness of the HyperCore.

**Strengths:**

1. The motivation is clear. The paper correctly identifies the need for a method that is both noise-robust and adaptive, which is a critical gap in the current literature.

2. The use of hypersphere-based class embeddings combined with Youden's J statistic for adaptive thresholding is an elegant idea with solid theoretical grounding. It effectively reduces heuristic tuning.

3. The paper is generally clear and easy to follow, with well-organized technical sections.

**Weaknesses:**

(Major)

The experimental evidence is not strong enough to substantiate the claims of robustness and generalization. Critical evaluations are missing:

1. No comparison with Random selection, which is a standard and strong baseline in DeepCore[Ref 1]).

2. While the paper clained that HyperCore raises the bar for robust coreset selection, setting new benchmarks for pruning accuracy and label noise tolerance, they only tested TyperCore on CIFAR-10 with 10% label noise. This paper should test:

(1)  on additional datasets, such as CIFAR-100 and ImageNet-1K,

(2) under different noise level, such as 20%, 40%, and 80% symmetric noise and 20% and 40% asymmetric noise,

(3) against recent robust coreset selection methods, such as [Ref 2-4].

3. The static pruning algorithm (Sec. 3.2) is not analyzed. Its standalone performance should be reported.

4. Table 3: The meaning of red and green results is unclear.

5. Table 4: The corresponding experimental settings are not described.

(Minor)

1. Citation error: Lines 33-34 "... informative subset that preserves the performance of training on the full dataset (?Sorscher et al., 2022 ..." (misformatted reference)

2. Some figures could benefit from improved captions and descriptions to clarify the setup.

More comments that do not impact the score:

- The method's core principle is to select the most "central" or "prototypical" samples (those with the smallest norm). As discussed by the authors in the limitations, and as is common with centroid-based methods/prototype selection methods, this might systematically discard valuable, informative outliers or samples that lie on the decision boundary, potentially limiting samples diversity, and this also ignores the distribution diversity. This could be discussed more.


[Ref 1] Guo C, Zhao B, Bai Y. Deepcore: A comprehensive library for coreset selection in deep learning[C]//International Conference on Database and Expert Systems Applications. Cham: Springer International Publishing, 2022: 181-195.

[Ref 2] Xia X, Liu J, Yu J, et al. Moderate coreset: A universal method of data selection for real-world data-efficient deep learning[C]//The Eleventh International Conference on Learning Representations. 2022.

[Ref 3] Xia X, Liu J, Zhang S, et al. Refined coreset selection: Towards minimal coreset size under model performance constraints[C]. ICML2024.

[Ref 4] Mohanty S, Anudeep C, Mopuri K R. Noise-free Loss Gradients: A Surprisingly Effective Baseline for Coreset Selection[J]. Transactions on Machine Learning Research, 2025.

**Questions:**

1. Could the authors provide additional experiments on CIFAR-100 or ImageNet-1K to demonstrate scalability and robustness under varying noise conditions?

2. Please clarify the baseline setup in Table 4 and define what "static pruning" performance refers to.

---

### Official Review · Reviewer_5eni · 2025-11-01

**Soundness:** 2
**Presentation:** 2
**Contribution:** 2
**Rating:** 2
**Confidence:** 3

**Summary:**

- The paper introduces coreset selection for dataset with annotation errors/noisy environments.
- The proposed approach depends on lightweight hypersphere models learned per class, embedding in-class samples close a hypersphere center while segregating out of class samples based on their distance.
- In more details, the method trains lightweight hypersphere models that learn to separate in-class from out-of-class (including erroneous labeled samples). It’s a process of separating samples that appear atypical when measured against a given class distribution, which are treated as outliers. Measure the distance of each point to its class-specific hypersphere centre and obtain an interpretable conformity score.
- J statistics is used to decide the hypersphere’s decision boundary.

**Strengths:**

1. Using a hypersphere is a straightforward concept and is a variation of Moderate [A] where samples lying in the median between class centres are picked as the coreset. The formulation depends upon the intuition that mislabelled samples lie farther from class centres in embedding space.
2. Expensive gradient and influence-based computations are avoided.
3. The use of adaptive threshold selection through Youden’s J statistics handles class imbalance.

**Weaknesses:**

**1. Missing baselines:**

Several important coreset selection baselines are missing. All the methods compared are part of DeepCore which was released in 2022 and there have been some influential works published in the field of coreset selection in the past 3 years.
A) Moderate Coreset: A Universal Method of Data Selection for Real-world Data-efficient Deep Learning (ICLR 2023)
B) Robust Data Pruning under Label Noise via Maximizing Re-labeling Accuracy (NeurIPS, 2023)
C) Coverage-Centric Coreset Selection for High Pruning Rates (ICLR, 2023)
D) Data Pruning via Moving-one-Sample-out (NeurIPS 2023)
E) Noise-free Loss Gradients: A Surprisingly Effective Baseline for Coreset Selection (TMLR, 2025)

**2. Limited Experiments**

Results are provided only for CIFAR-10 and ImageNet-1K, while other standard datasets such as CIFAR-100 and Tiny ImageNet are not utilized. These would have provided insight into how the methodology behaves with varying number of classes.

**3. Timing comparison**

Timing comparison seems misleading, as all the other methods being compared , select coreset for the entire dataset, while the proposed method selects coreset in a class wise manner. Assumption that sufficient number of parallel cores and GPUs are available defeats the core intention of coreset selection i.e. efficient deep learning method in a compute-constrained environment.
Timing analysis is provided only for CIFAR-10 dataset. The scale up required for 1000 class dataset of ImageNet-1K would definitely be very time intensive.

**4. Performance of the method**

As reported in Table 1 and Table 2, the proposed method barely outperforms existing methods for only a few of the selection fraction. The trade-off of compute time due to the nature of per-class selection against very minimal performance gain (for a very few selected fractions) seems trivial.

**5. Dataset not unexplored**

As the paper claims that the method is suitable specifically in cases of datasets with label noises, it would have been helpful to understand the performance of the method on datasets such as CIFAR-100N and CIFAR-10N, which are datasets with realistic label noise due to human annotation error.

**6.  Lack of theoretical and empirical analysis of hyperspheres for coreset selection**

The paper does not provide any supporting proof towards hyperspheres being able to select representative samples as compared to samples with label noises. An analysis of percentage of clean samples and percentage of noisy samples being selected in the coreset would be helpful.

**7. Alternate thresholding criteria [MINOR]**

The paper does not explore (or atleast discuss) alternative thresholding criteria other than the J statistics. An ablation of alternate thresholding criteria would help in highlighting the impact of J statistics.

**8. Formatting issues [MINOR]**

There is a “?” mark in line 034, which seems to be a missing reference.

**Questions:**

- Please refer to the weaknesses section of this review.

---

### Official Review · Reviewer_Y7bx · 2025-11-03

**Soundness:** 3
**Presentation:** 3
**Contribution:** 2
**Rating:** 2
**Confidence:** 4

**Summary:**

HyperCore introduces a coreset selection method leveraging class-wise hypersphere models, which effectively identify and retain representative data samples under noisy conditions. Assuming the data are fully labels, hypercore trains per-class models and then adapts a per-class threshold with Youden’s J statistic.  It seems able to efficiently discards mislabeled or outlier points without manual parameter tuning. The approach is analyzed on CIFAR-10 and on ImageNet-1K.  A compelling result that shows robustness to noise in the data is alao included.

**Strengths:**

- S1: The notion of adapting parameters per class (if you have the labels) is interesting and relevant.  And the use of classical signal detection theory ideas to adapt each threshold is clear.

- S2: Likewise explicitly capturing noise in the data and analyzing robustness to such noise is a nice addition to the coreset literature.

- S3: The method is relatively simple leading to what seem to be low computational costs, although no comparative wallclock time is given, the networks trained are "small."

- S4: The exposition is clear and the paper is understandable.

**Weaknesses:**

- W1: The basic assumption in the paper --- that this is a fully supervised problem with a label for all samples --- is barely tacitly stated (L078).  Yet, this problem setup differs from the original idea in Sener and Saverese 2017 which is an active learning setup where only a subset of the labels exist at any iteration and the goal is to find the next subset of unlabeled samples to label.  Although numerous papers in the coreset literature make this same questionable deviation from the original problem, it is highly questionable.  The motivation in the introduction does not use the motivation from Sener and Saverese (namely that labeling very large datasets is prohibilitively expensive and complicated), but rather shifts to the notion that it is possible to get better performance from a best core subset.  This shift renders the problem mostly irrelevant from an impact standpoint.

- W2: Critical competitive references are not included -- e.g., Temporal Dual-Depth Scoring Zhang et al. CVPR 2024.  In fact, no comparative result in the tables exist for papers after 2021???

- W3: Coreset selection is a challenging problem (with or without class labels).  Interestingly, reports (e.g., https://arxiv.org/pdf/2411.15349) have shown that uniform sampling is a very competitive baseline.  This paper has no such analysis against the competitive baseline.  EG. Looking at the prune rate of 50% for CIFAR10 in that paper and this paper (both using ResNet18 as the downstream detector), the uniform sampler outperforms this HyperCore (AND the uniform sampler uses no class labels).

- W4: The static pruning baseline is derived in Sec 3.2. But it is not included in any of the tables.


Other things
- L033 -- Missing Reference?

**Questions:**

- Q1: Although the hypershere model seems useful as it is relatively straighforward to fit, one wonders whether an even slightly more complex model that can adapt to each dimension would be a better performer (although increase fitting complexity Eq 2).  In these high dimensions
- Q2: Want to confirm that $\mathbf{x}$ is always the raw input and not some embedded form of it.
- Q3: What is the actual comparative wall-clock time of training these small models?  What is the size of these models?

**Details Of Ethics Concerns:**

Probably this is not a true ethics concern.  But, it is very clear to me that 13521 and 2361 were contributed by the same authors.  The problem statements are the same, chunks of text are common, and the results have the same limitations.  The methods seem to be sufficiently different (although in the end all they are really doing is fitting local densities to data-samples and the sampling from those densities).

---

### Official Review · Reviewer_ev9C · 2025-11-04

**Soundness:** 2
**Presentation:** 3
**Contribution:** 2
**Rating:** 6
**Confidence:** 3

**Summary:**

The paper provides a subset selection algorithm for data especially when there is noise in the data in form of labelling errors. Their algorithm learns a hypersphere for every class separately and uses the distance of every point to the center of the hypersphere to decide if the sample should be in the subset. The distance is compared with an adaptive threshold determined automatically using Youden’s J statistic. Extensive empirical evaluations show the effectiveness and scalability of their method especially for noisy environments.

**Strengths:**

1) The paper is well written and is easy to follow. The problem of subset selection for training ML models is a very important problem and of interest to community.
2) The algorithm is intuitive, and its parallelizability makes it very efficient
3) Empirically the algorithm is compared with a bunch of baselines in terms of time and scalability both in noisy and noise free settings and the results are encouraging.

**Weaknesses:**

I appreciate that the authors have listed out most limitations of their method. However, one key limitation is that the method looks more heuristic in nature. some of the baselines while performing slightly weaker empirically do come with guarantees about the quality of sampling.

The experiments are compared with a variety of baselines, however, as pointed out in the point above, since this is less theoretically strong method compared to few other baselines, it would be useful to test on a couple of other datasets/ downstream tasks too.

Question: Elaborate on the effect of using $\mathbf{c =0} $ as center. Does it change quality of hypersphere obtained or effect sample size other than making the method computationally efficient?

**Questions:**

See Weaknesses

---

### Note · Authors · 2025-11-16

I have read and agree with the venue's withdrawal policy on behalf of myself and my co-authors.